# Investigation and Analysis of Rhizosphere Soil of Bayberry-Decline-Disease Plants in China

**DOI:** 10.3390/plants11233394

**Published:** 2022-12-06

**Authors:** Gang Li, Jingjing Liu, Yu Tian, Han Chen, Haiying Ren

**Affiliations:** 1Institute of Agro-Product Safety and Nutrition, Zhejiang Academy of Agricultural Sciences, Hangzhou 310021, China; 2College of Environmental and Resource Sciences, Zhejiang University, Hangzhou 310058, China; 3Institute of Horticulture, Zhejiang Academy of Agricultural Sciences, Hangzhou 310021, China

**Keywords:** bayberry decline disease, soil parameters, aluminum toxicity, soil acidification

## Abstract

The rampant bayberry decline disease has been regarded as related to soil with the long-term plantation bayberry. These parameters, hydrogen, aluminum, other alkali cations, and plant-related nutrients, were measured from the soil around diseased tree roots 10, 20, and 30 years old. The pH significantly declined in topsoil with increasing tree age and rose with increasing depth of the soil layer with an age of 10, 20, and 30 years. The concentration of exchangeable aluminum has risen significantly with the increase of the tree ages in the top soil layer and also in 0 to 40 cm soils layer with ten-year-old trees. In the top soil layer with a depth of 0 to 10 cm, the cation concentrations of Ca^2+^, Mg^2+^, and K^+^ has fallen significantly with the increase of tree ages. A higher concentration of exchangeable aluminum was observed in the soil with trees more seriously affected by the disease and was accompanied with lower concentrations of Ca^2+^, Mg^2+^, and K^+^. The correlation analysis showed that the soil pH is significantly positively related to the concentration of exchangeable Ca^2+^, total nitrogen, and total phosphorus and negatively to exchangeable aluminum. These findings provided a new insight to mitigate the disease by regulating the soil parameters.

## 1. Introduction

Bayberry (Myrica rubra) is an important economic and ecological tree species in major bayberry-producing areas of China, and its planting area has been rapidly expanding for its comprehensive economic benefits in recent decades. As local fruit trees in China, it is an evergreen with luxuriant branches and leaves, vigorous growth, long economic life, and strong adaptability. However, in recent years, adult bayberry trees in major production areas in Zhejiang Province have suffered from leaves falling off, small fruit with poor quality, and death after occurrence in 3–5 years, which is named the “decline disease” [1,2].

Its pathogenic mechanism remains unclear. No evidence indicated that the disease was caused by a pathogenic microbe. Multisite field surveys showed that the occurrence of the disease was closely related to soil. Rhizosphere soil of diseased trees compared healthy trees vary in contents of pH, organic matter, magnesium, available phosphorus, nitrogen, and calcium [2,3]. Soil with improper parameters about plant nutrients may be partially responsible for the occurrence of bayberry decline disease. The disease could be alleviated in the condition of improved rhizosphere soil with the optimization of physical and chemical properties and microbial community by application of fertilizer. For example, the application of bio-organic fertilizer in soil exhibited an enhancement effect of bayberry plant health and soil quality by regulation of soil microflora, such as the increase of the relative abundance of *Burkholderia*, *Mortierella,* and *Geminibasidium* and the decrease of that of *Rhizomicrobium, Acidibacter, Trichoderma* and *Cladophialophora* [3].

Improper long-term management of the bayberry tree may lead to a change of the soil parameters around the tree roots. Such as for the increase of fruit yields, a large amount of chemical fertilizer (artificial NPK) is applied in bayberry orchards, while the organic manures (stable manure or composted plant) are not for labor savings, which resulted in a change of the soil parameters in bayberry orchards in Southern China. For example, the average soil pH value in typical adult bayberry orchards of Zhejiang Province was 4.5 for now and 5.9 for 30 years ago, as a result of the combined effect of its topography of hilly and mountainous areas with acid rain and excessive chemical fertilizer employment [4]. The public data showed that bayberry decline disease was attributed to soil acidification, and it results in a high concentration of exchangeable aluminum, which is of a toxicity to plants for directly inhibiting and damaging of root tissues and causing deficiencies in Ca, Mg, and P, as well as other physiological stresses, and which decreased the bacterial diversity and altered the bacterial community structure, such as a decrease of the relative abundance of *Acidobacteria* and *Planctomycetes*, and increase of that of *Cyanobacteria*, *Bacteroidetes*, and *Firmicutes* [5]. However, soil microbes are the essential factors for plant growth and development and for enhancement of the resistance of plants to disease by the improvement of soil fertility and the decomposition of organic matter and the addition of beneficial bacteria [6]. In addition, aluminum in the solid phase exists in four forms, exchangeable aluminum (ExAl), monomeric hydroxyl aluminum (HyAl), colloidal aluminum (CoAl), and humic acid aluminum (HaAl). The increased soil acidity would greatly elevate the concentration of labile Al^3+^, which has been reported to cause the direct inhibition of root elongation and interfere with the absorption of plant nutrients in soil, which is also known as plant aluminum toxicity [7,8,9].

Currently, there is no precise and effective single control method for the disease due to the unclear pathogenic mechanism. However, integrated control measures can provide reliable and multiple effects to control the disease and rejuvenate the trees in a quasi-health state for fruit yields. The comprehensive disease management practices include a chemical control to wipe out the pathogen coupled with a physical control to decrease the primary inoculum in the spring and coupled with organic fertilizer to amend the soil and make the tree vigorous. In the chemical control, a water solution with one synthetic fungicide, pyraclostrobine, prochloraz, difenoconazole, or azoxystrobin, was yearly alternately sprinkled on ploughed rhizosphere soil after harvest in July or August. In the physical control, sanitation was applied to decrease the primary inoculum by removing diseased trees and its debris out of a bayberry orchard in the spring coupled with pruning and covering the wound with copper sulfate. In the organic fertilizer control, biochar-based fertilizer, bio-organic fertilizer, or animal-original manure such as fermented sheep dung stable manure and fermented shredded disease trees as the main ingredient in manure were applied for nutrient supply and soil amendment [10].

In this study, we focused on a survey of soil parameters in bayberry orchards in Zhejiang Province, the largest bayberry cultivation zones in China. Our objectives were to discover the characteristics of the soil parameters in bayberry decline disease orchards to explore possible relations between the change of soil parameters and the disease for establishing a soil-based and environment-friendly control method of the disease.

## 2. Results

### 2.1. Disease

In ten study sites, bayberry decline disease incidence and its severity (disease index) on deciduous leaves ranged from 16.3 to 42.3% and from 12.5 to 30.2 for 10-year-old trees, from 26.4 to 51.7% and from 18.5 to 45.2 for 20-year-old trees, and from 35.0 to 59.5% and from 27.9 to 55.2 for 30-year-old trees, respectively (Table 1). While bayberry decline disease incidence and its severity (disease index) on single fruit weight ranged from 14.0 to 40.3% and from 15.9 to 29.6 for 10-year-old trees, from 27.0 to 52.3% and from 11.3 to 37.6 for 20-year-old trees, and from 37.0 to 65.3% and from 17.3 to 43.6 for 30-year-old trees, respectively.

In each study site, there were similar rising trends of incidence and its severity on defoliation and single fruit weights with the increasing of tree ages. Bayberry decline disease incidence and its severity (disease index) on deciduous leaves ranged from 16.3 to 52.3% and from 12.5 to 55.2 for the increase of tree ages from 10 to 30 years. Bayberry decline disease incidence and its severity (disease index) on single fruit weight ranged from 14.0 to 65.3% and from 11.3 to 43.6 for the increase of tree ages from 10 to 30 years.

### 2.2. pH and Exchangeable Acidity in Soil

The soil pH and the concentration of exchangeable acidity were prominently different with the increase of diseased tree ages in the soil of study site and no obvious pattern in that of healthy trees. In the horizontal direction of the soil layer around disease tree roots, its pH declined with increasing tree ages from 10 years old to 20 or 30 years old in the above two layers (from 0 cm to 10 cm and 10 to 20 cm). In the perpendicular direction of the soil layer, its pH has risen around the 20 and 30-year-old tree roots with increasing depths and fell around 10-year-old tree roots with increasing depths (Figure 1a). On the other hand, in the horizontal direction of the soil layer around disease trees, its concentration of exchangeable acidity has significantly risen with increasing tree ages in the above two layers. In the perpendicular direction of soil, its concentration of exchangeable acidity has significantly risen around the 10-year-old trees with increasing depths and has no significant changes around 20 and 30 year-old trees roots with increasing depths (Figure 1b).

### 2.3. Aluminum in the Soils

In each soil sample, the concentrations of the aluminum forms decreased in the order of HaAl > CoAl > ExAl > HyAl. In the horizontal direction of the soil layer around disease tree roots, the concentrations of aluminum have significantly risen with increasing tree ages in the above two layers for ExAl and HyAl and no change for HaAl and CoAl in each layer. While in the perpendicular direction of the soil layer, the concentration of aluminum has significantly risen with increasing depths for HyAl and ExAl from the soil around the tree roots 10 years old and no significant changes for HaAl and CoAl (Figure 2). The Pearson correlation analysis showed that a significant negative correlation was found between the soil exchangeable aluminum and soil pH (*R*^2^ = 0.89, *p* < 0.01) (Figure 3).

### 2.4. Exchangeable Cations in Soil

Long-term bayberry plantations have effects on the concentration of exchangeable cations Ca^2+^, Mg^2+^, and K^+^ in the soil around bayberry tree roots. In the horizontal direction of the soil layer around disease tree roots, the concentrations of exchangeable cations Ca^2+^, Mg^2+^, and K^+^ have significantly declined with increasing tree ages in the top soil layers and also above the second soil layers for Ca^2+^ and no changes for Ca^2+^, Mg^2+^, and K^+^ in the other three layers of soil, except for Ca^2+^ above the second soil layers. While in the perpendicular direction of soil layer, the concentrations of exchangeable cations Ca^2+^, Mg^2+^ and K^+^ have significantly risen with increasing the depth in the soil around the tree roots to 20 and 30 years old and no significant changes for 10-year-old trees (Figure 4). The Pearson correlation analysis showed that a significant positive correlation (*R*^2^ = 0.77 *p* < 0.01) was found between soil exchangeable Ca^2+^ and soil pH (*p* < 0.01) (Figure 5).

### 2.5. The Concentration Correlation of Cations (H^+^ and Aluminum) with Nutrients in Soil

The correlation between four soil acidification indicators (including pH, exchangeable acidity, exchangeable aluminum, and hydrolytic acidity) and nine soil nutrient factors was analyzed (Table 2). The available phosphorus was significantly negatively correlated with exchangeable acidity and exchangeable aluminum at *p* < 0.05 and no correlated with pH and hydrolytic acid. The total phosphorus was significantly positively correlated with the pH and negatively with the exchangeable acidity at *p* < 0.001. The total nitrogen was significantly negatively correlated with the exchangeable acidity and exchangeable aluminum (*p* < 0.05) and positively correlated with the pH. The relationship between the available nitrogen and acidification indicators is not significantly correlated.

### 2.6. Correlation Analysis between Acidity—Related Cations and Nutrients in Soil

Structural equation modeling (SEM) was used to discover causal relation between acidity–related cations and nutrients in the soil. The model included observable variables that are directly measured and potential variables that are used to describe variables unobserved. Three potential variables, the acidification index (AI), aluminum forms index (AL), and nutrient index (NI), were applied in the model. The fitness of the model was evaluated by the chi-square degree of freedom ratio (CMIN/DF), goodness of fit index (GFI), comparative fit index (CFI), normed fit index (NFI), and root mean square error of approximation (RMSEA). The final fitness parameters indicated that the model had high credibility (RMESA = 0.098, CFI = 0.937, NNFI = 0.918, IFI = 0.941, X^2^/df = 1.3). The SEM indicated that the AI in soils was obviously affected by AL and NI, evidenced by the path coefficients of −0.77 for AL and 0.66 for NI.

All nutrient indexes (NI) have affected the total acidification index (AI) by influencing the potential variable instead of the observable variables in a single path. Soil organic matter (SOM) did not play an important role in determining the value of the nutrient index (NI) with a lesser coefficient of 0.2, while the exchangeable calcium cation Ca^2+^ showed a significant effect on the value of the nutrient index (NI) with the path coefficient of 0.99. The total nitrogen, available phosphorus, and alkali hydrolyzed nitrogen had certain effects on the nutrient index (NI), but such effects were relatively slighter compared to that of the calcium cation Ca^2+^ (Figure 6).

The soil pH, hydrolysable acid, and exchangeable acid had strong correlations with the acidification index (AI). The potential variable acidification index (AI) was directly significantly affected by the single factor with a high coefficient (Figure 6). On the contrary, similar to NI, which had a coefficient (0.66) with AI, the high potential variable coefficient with low single factors showed that the acidification index (AI) was driven by the potential variable nutrient index (NI). After accounting for all effects, the soil acidity condition was significantly associated with all the single acidification factors, aluminum forms, and the nutrient index, especially the calcium cation Ca^2+^ and exchanged aluminum. The direct regression analysis between the calcium cation and pH indicated that calcium had a strong positive correlation with the pH (Figure 5). There was no direct path coefficient between calcium and the single acidification factors in the model (Figure 6), but the calcium affected the acidification index through the potential variable nutrient index (NI), which had a high coefficient (0.66) with the acidification index (AI).

Considering the correlation (−0.77) between the aluminum forms index (AL) and acidification index (AI) and the high coefficient (0.98) between the exchangeable aluminum and aluminum forms index (AL), the exchangeable aluminum affected the soil acidification condition not only by the individual factor but also by the whole potential path ways. HyAl also played a role in the impact of the form of aluminum on the soil acidification condition, since it had a negative path coefficient (−0.25) with aluminum, which could not be ignored. Both monomeric hydroxyl aluminum and exchangeable aluminum were variables measured of the same potential variable (aluminum forms index, AL), and there might be a coupled influence between them.

## 3. Discussion

### 3.1. The Changes of Acidity-Related Parameters in Soil

The survey results showed that there are prominent changes of the acidity-related parameters, including pH and exchangeable acidity in the soil. The pH significantly declined in the topsoil (0–20 cm) with the increasing tree ages and rose with the increasing soil depth from 20 cm to 40 cm around the tree roots aged 20 or 30 years old. In the survey orchards, the fertilizer management is mainly dependent on the input of chemical fertilizer (artificial NPK) for lasting over fifteen years. Therefore, inputting large amounts of chemical fertilizer in bayberry orchards is an important driving factor for soil acidification. The optimal soil pH for bayberry trees ranges from 5.5 to 6.5, with 5.5 being the preferred pH. Previous reports also indicated that soil acidification was observed in bayberry orchards, such as the pH investigated in Cixi Country of Zhejiang Province in 1985, 2012, and 2014 being 5.86, 4.55, and 4.51, respectively [4,11,12]. The trend of soil acidification is growing more and more serious in bayberry orchards of Zhejiang Province with the increase of tree ages. Soil acidification is harmful to some plants, especially to cereal crops, forests, and grasslands [11,12,13,14,15,16]. Our survey also showed that higher frequencies of pH 4.5 in the soil samples were observed, which may have adverse effects on bayberry growth and the occurrence of bayberry decline disease. When the soil pH is lower than 4.5, bayberry tree growth is inhibited, affecting both the quality and quantity of the fruit [17,18]. Chen H. et al. (2022) reported that soil acidification is a cause for bayberry decline disease.

Soil acidification is a natural process in which non-acid cations are leached and H^+^ increases. Agricultural activities such as overdose using a chemical fertilizer can accelerate this process [19]. The following reasons maybe responsible for the decrease of pH in the soil: (i) the overdose use of chemical fertilizer (artificial NPK). Single-use chemical fertilizer and the lack of traditional agricultural measures is proven to be responsible for the soil acidification in tea orchards [16,20,21,22]. (ii) The continuous loss of soil base cations (Ca^2+^, Mg^2+^, and K^+^) induced by fruit harvesting leads to soil acidification [12,23,24,25], and (iii) bayberry trees were mainly planted in subtropical mountain region with large precipitation, strong and intensive leaching, and acid deposition leads to soil acidification [15,19,25].

Soil acidification could be ameliorated by appropriate amounts of quicklime, calcium magnesium phosphate fertilizer, and organic fertilizer, which are good soil remediation agents to improve the soil pH value, alleviate plant diseases, and effectively keep the plant healthy by balancing the soil nutrients, which plays an important role in plant growth and development [26].

### 3.2. The Change of the Concentration of Aluminum in Bayberry Orchard Soil

Soil acidification has become a global concern and not only leads to plant disease and decreases the biodiversity of ecosystems but also causes soil nutrient losses, such as soil acidification inducing soil exchangeable Al and available Cu to increase and exchangeable cations to decrease in rubber plantations [19]. In this investigation, we observed that, with the increase of the tree ages, the concentration of exchangeable aluminum has risen significantly in the top soil layer from 0 to 20 cm around diseased tree roots and also in soil layers from 0 to 40 cm around diseased tree roots ten years old.

Aluminum species in the soil are determined by the pH [27,28]. The exchangeable aluminum concentrations that are toxic to plants were significantly negatively correlated with the pH in a wide variety of soils [25,29]. As the soil concentration of H^+^ increased, and monomeric hydroxyl aluminum (such as Al(OH)^2+^ and Al(OH)_2_^+^) was gradually transformed into Al^3+^, and when the soil pH falls below 5.5, a large amount of soluble ionic aluminum will be released into soils. Therefore, aluminum is considered to be phytotoxic at soil pH below 5.5 [30]. In this survey, the soil’s planted bayberry for 30 years had significantly lower pH and higher levels of exchangeable aluminum compared to that in 10- and 20-year-old trees. Therefore, aluminum toxicity may be responsible for the bayberry decline disease and relieved by regulating the soil pH. Beyond that, the large amounts of exchangeable Al released in bayberry orchards is a threat to the health of the surrounding environment due to the effects of runoff and soil erosion.

In addition, the degree of aluminum toxicity also depends on other factors, including the clay minerals, organic matter content, concentrations of other cations, anions, total salts, and the type of plant species present [5]. In the surveyed bayberry orchard, the organic matter content, which is poor, could generally be related to a high concentration of aluminum due to its weak absorption into aluminum in high pH.

What is more, more attention should be paid to other metal cations, including weighted metals, which also have adverse effects on the plants and environment.

### 3.3. The Change of Plant-Related Nutrient in Bayberry Orchard Soil

Soil acidification enhances the leaching of exchangeable cations of Ca^2+^, Mg^2+^, Zn^2+^, Mn^2+^, and Fe^2+^ [19,31]. In this survey, we observed that, in the top soil layer with a depth of 0 to 10 cm, the base cation concentrations of Ca^2+^, Mg^2+^, and K^+^ have fallen significantly with the increase of tree ages, and with the increase of soil depth, the concentrations of Mg^2+^ and K^+^ have fallen significantly more in the soil around 30-year-old disease tree roots than in that of 10- and 20-year-old disease tree roots. Deficiency of the cations’ nutrition in the soil may relate to the disease.

By the correlation analysis, the soil pH is significantly positively related to the exchangeable Ca^2+^, total nitrogen, and total phosphorus and negatively to the exchangeable aluminum. Therefore, it is easy to obtain the following instructive inference: (1) soil acidification can accelerate the loss of soil nutrients (e.g., extractable calcium, magnesium, and potassium) and elevate the concentrations of toxic aluminum in the soil. (2) The deficiency of the nutrients, particularly calcium, and aluminum toxicity could be one adverse factor of occurrence of the decline disease or yield reduction in bayberries.

In addition, loss of the calcium ions in acidic soil might be attributed to the following causes: (1) the calcium ions were easily leached and migrated out of the soil with rainfall for not being adsorbed by the soil colloid in high concentrations of the hydrogen ion exchange system, (2) and worse, the base cations, including the calcium ions, were dwindling fast when the concentration of exchangeable aluminum increases exponentially in the soil with a lower pH, because the dissolved aluminum has a higher adsorption affinity to colloidal particles than that of the base cation in acidic soil.

It could be deduced that the exchangeable aluminum resulted in lower mineral nutrition and poor physiological metabolism of bayberry trees. Therefore, it may be a good perspective for controlling the bayberry decline disease by regulation of the balance of these soil parameters.

## 4. Materials and Methods

### 4.1. Study Sites

The study sites were located hill-mountain areas in Zhejiang province, eastern China, which is the major bayberry plantation area in China. In this region, bayberry has become the most important economic fruit. The study sites lie in humid mid-subtropical monsoon climate, with the mean annual temperature of 18.4 °C and the mean annual precipitation of approximately 1560 mm. A bayberry orchard as a study site is under traditionally agricultural management as the usual employment of chemical compound fertilizers (main nutrients items including nitrogen, phosphate, and potassium) and almost no other organic fertilizers. There are totally 10 study sites randomly chosen in this study which are distributed in ten different counties randomly chosen, including Shangyu, Cixi, Dinghai, Lanxi, Qingtian, Xianju, Linhai, Tiantai, Taishun, and Wencheng. They had similar altitude, landform, soil condition, and initial soil fertility. These study sites originally covered by wild nature forest, which is mainly composed of subtropical evergreen, deciduous broad-leaved forests before bayberry plantation. The soils are derived from rock broken down into dirt (mainly sandy rock) and classified as Alfisol [32].

### 4.2. Soil Sampling

In each study site, including different ages trees in every orchard; the soil under the canopy of healthy and diseased trees was randomly collected as sample. These trees are in different tree age of 10, 20, and 30 years old, which were marked as T10, T20, and T30, respectively. The tree age is identified by plantation recording provided by the farmers’ own of the trees. The diseased trees with different age are randomly distributed in the bayberry orchards. The old tree possesses a larger canopy and tree trunk than the young one. The proportions of diseased trees are no more than 70% in the all surveyed orchards. Each sampling spot was vertically stratified and sampled from top surface of 0–10 cm, 10–20 cm, 20–30 cm, and 30–40 cm. The soils around three to five randomly chosen the same age trees roots were sampled in similar weight to form a spot sample. The five spot samples were mixed to form a study site sample which was completed using method of five sampling spots. The soil sample was placed in a plastic bag and transported to the laboratory. Soil samples were air-dried at room temperature and cleaned by discarding stones, roots, litter and other plant debris and sifting and collecting with a 2-mm sieve and stored in plastic self-close bags for next analyses.

### 4.3. Disease Assessments

Bayberry decline disease was assessed for incidence and its severity (disease index) on defoliation and single fruit weight loss. In 2021, deciduous leaves and all leaves from each quadrant of disease and healthy trees randomly selected with different ages of 10, 20, and 30 years old were counted on December in all ten study sites, and samples of mature fruit were collected from the same trees in June. The deciduous leaves were investigated from the ten trees as a replicate with 3 repeats per tree age. Fifty fruits as a replicate sample were collected from the same trees with 3 repeats per tree age. Incidence for defoliation was calculated as the percentage of deciduous leaves on total leaves of each quadrant of a tree, and incidence for single fruit weight as the percentage of fruit weight from disease trees on that of healthy trees. Incidence was classified into nine classification grades described as intervals around the following values of defoliation and fruit weight percentages: 0% (grade 1), 1% (grade 2), 3% (grade 3), 12.5% (grade 4), 25% (grade 5), 37.5% (grade 6), 50% (grade 7), 75% (grade 8) and 90% (grade 9). The disease index for defoliation and single fruit weight was calculated with the formula: Σ (grade × numbers of trees with the same grade) × 100/total numbers of trees investigated.

### 4.4. Measurement of Soil pH and Exchangeable Acidity

Soil pH was measured using a soil to solution ratio of 1:2.5 (*w*/*v*) with a pH meter (Metler Toledo, Shanghai, China), while exchangeable acidity (EA) of the soil was extracted with 1 M KCl solution and determined by titration with 0.1 M NaOH [33].

### 4.5. Measurement of Soil Nitrogen and Phosphorus

The basic soil properties were determined according to the routine procedure [33]. Soil total carbon (TC) and total nitrogen (TN) were determined by an elemental analyzer (Flash EA1112, Thermo Finnigan, Italy). Total phosphorus (TP) was determined by digesting the 0.2 g of soil samples with sodium hydroxide and Mo Sb anti-spectrophotometrics. Soil available phosphorus (AP) was extracted by adding 20 mL of 0.5 M NaHCO_3_ (pH 8.5) to 1.0 g of soil, shaking for 30 min, and filtering through a filter paper. The concentration of P was measured by molybdenum blue colorimetry. Soil available nitrogen was determined by the alkali solution diffusion method. The composition of major elements in soils was determined by X-ray fluorescence spectrometer (XRF) (S8 TIGER, Bruker AXS, Germany).

### 4.6. Measurement of Aluminum Forms in Soils

Different forms of aluminum in the soil solid phase were sequentially extracted with 2 g soil samples using different reagents [34,35]. Exchangeable Aluminum was determined using aluminum colorimetry. Briefly, 20 mL of 1.0 M KCl, 20 mL of 1.0 M NH_4_OAc (pH 4.0), 20 mL of 1 M HCl, and 20 mL of 0.5 M NaOH were used to extract soil exchangeable aluminum, monomeric hydroxyl aluminum, colloidal aluminum and humic-acid aluminum, respectively. The concentration of aluminum in the extracts was determined by spectrophotometric with chrome azurol S [36,37,38]. Exchangeable H was calculated as the difference between the exchangeable acidity and the exchangeable aluminum.

### 4.7. Measurement of Exchangeable Cations

Exchangeable cations (K^+^, Na^+^, Ca^2+^ and Mg^2+^) were extracted with 1 M NH_4_OAc (pH 7.0) solution. All samples were filtered through a 0.45 μm filter paper and the filtrates were analyzed for cations. Exchangeable Na^+^ and K^+^ were measured by flame photometry (novAA 300, Analytik Jena, Germany), and exchangeable Ca^2+^ and Mg^2+^ were measured using atomic absorption spectroscopy [39].

### 4.8. Data Analyses

A one-way analysis of variance (ANOVA) was used to test the effects of plantation year on soil parameters. Differences between means were tested using the least significance difference (LSD) at *p* = 0.05. All the statistical tests were performed using SPSS software (version 24.0, SPSS Inc., Chicago, IL, USA). Pearson correlation analyses were used to assess the relationships between soil acidity and other soil parameters using SPSS16.0 (IBM Corporation, Armonk, NY, USA). The structural equation model (SEM) was employed to identify the relationship network of soil acidification [40]. The three parameters, acidification condition, aluminum form, and nutrition which were directly measured, were potential variables in SEM method. The SEM was constructed using the maximum likelihood estimation method. Multiple indexes, including comparative fit index (CFI), root mean square error of approximation (RMSEA), goodness fit index (GFI), normed fit index (NFI) and chi-squared degree of freedom ratio (CMIN/DF), were used to evaluate model fitting. In R statistical software, the ‘lavaan’ package was used to build the SEM, and the ‘randomForest’ package was used to analyze and evaluate the important factors in different variables and the degree of mutual influence.

## 5. Conclusions

The plant growth-related soil parameters have changed with the increase of tree ages in bayberry decline disease orchards. The pH significantly declined 0 to 20 cm in the topsoil with increasing the tree age and rose with increasing the depth of the 20 cm to 40 cm soil layer with a tree age of 20 or 30 years old. The soils with trees ages of 30 years old had significantly lower pH and higher concentrations of exchangeable aluminum compared to that in 10 and 20 year old trees. The concentration of exchangeable aluminum has risen significantly with the increase of the tree ages in the top soil layer from 0 to 20 cm and also in the 0 to 40 cm soil layer with ten-year-old trees. In the top soil layer with a depth of 0 to 10 cm, the cation concentrations of Ca^2+^, Mg^2+^, and K^+^ have fallen significantly with the increase of tree ages. With the increase of the soil depth, the concentrations of Mg^2+^ and K^+^ have fallen significantly in the soil around 30-year-old disease tree roots. A higher concentration of exchangeable aluminum was observed in the soil with more serious disease trees and was accompanied by lower concentrations of Ca^2+^, Mg^2+^, and K^+^. Based on the correlation analysis, the soil pH is significantly positively related to the exchangeable Ca^2+^, total nitrogen, and total phosphorus and negatively to the exchangeable aluminum. Overall, these findings provided a new insight to mitigate bayberry decline disease by regulating the above-changed soil parameters.

## Figures and Tables

**Figure 1 plants-11-03394-f001:**
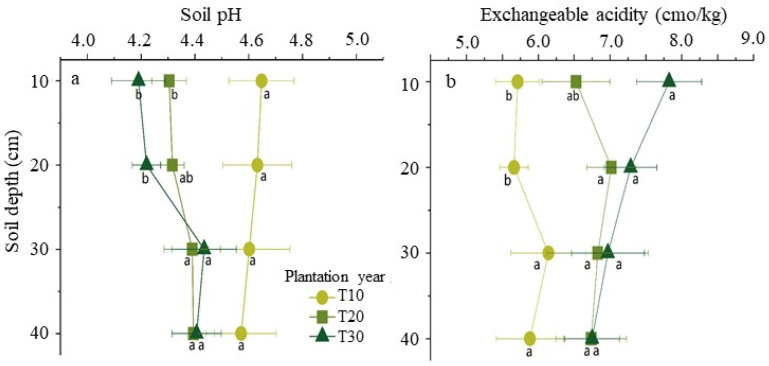
Temporal and spatial distributions of pH (**a**) and exchangeable acid (**b**) in the soil around disease bayberry tree roots. The different letters represent significant differences in different tree age bayberry (*p* < 0.05).

**Figure 2 plants-11-03394-f002:**
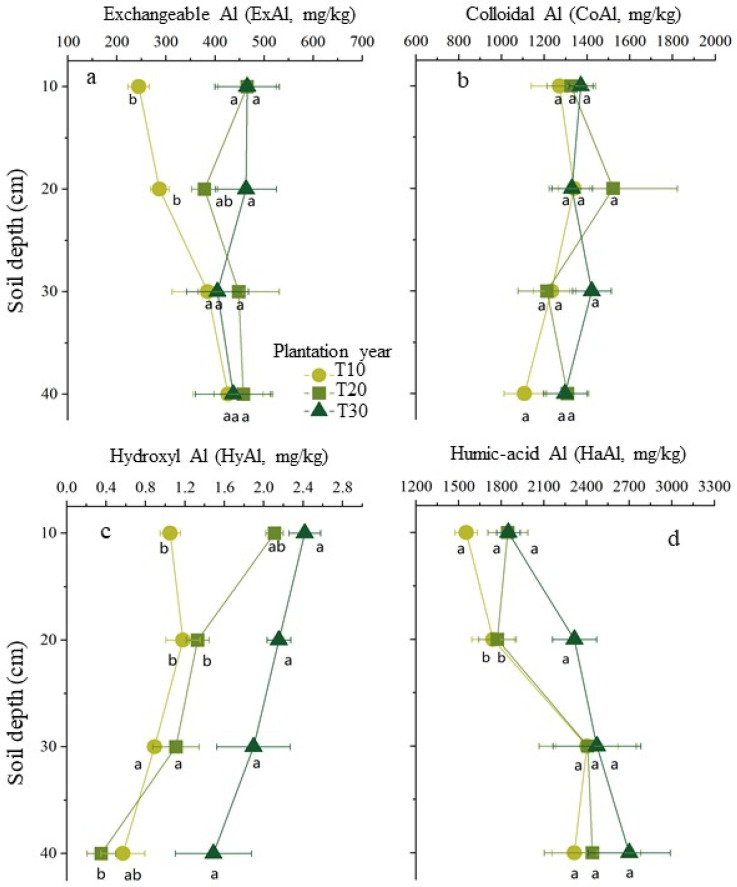
Temporal and spatial distribution of 4 main aluminum (Al) forms in soils around bayberry tree roots. The concentrations of aluminum (abscissa axis) in different soil depth (axis of ordinates) from sample soil of the different tree ages of T10 for 10 years old, T20 for 20 years old, and T30 for 30 years old were showed for exchangeable Al in (**a**), for colloidal Al in (**b**), for hydroxyl Al in (**c**), and for humic-acid Al in (**d**). The different letters represent significant differences in different tree age bayberry (*p* < 0.05).

**Figure 3 plants-11-03394-f003:**
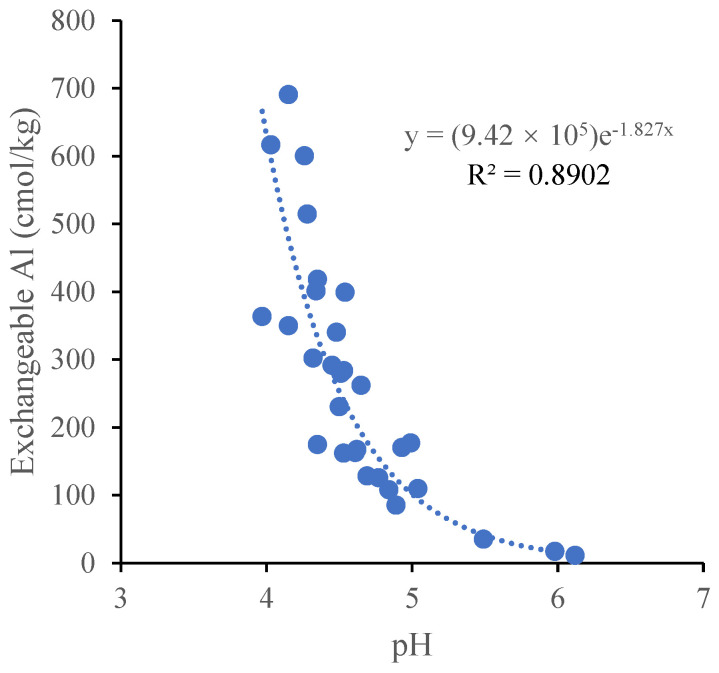
Relationship between pH and exchangeable aluminum in soils around disease bayberry roots. Each dot represents a soil sample in the figure.

**Figure 4 plants-11-03394-f004:**
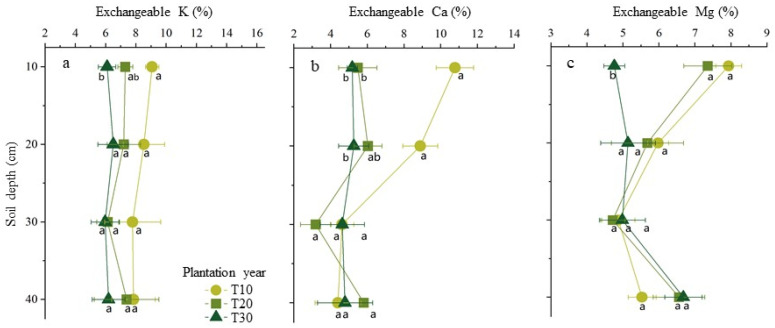
Temporal and spatial distributions of exchangeable Ca, Mg, and K in the soil of bayberry orchards. The contents of exchangeable Cations (abscissa axis) in different soil depth (axis of ordinates) from sample soil of the different tree ages of T10 for 10 years old, T20 for 20 years old, and T30 for 30 years old were showed for exchangeable K in (**a**), for exchangeable Ca in (**b**), and for exchangeable Mg in (**c**). Different letters between different bayberry tree ages represent significant differences (*p* < 0.05).

**Figure 5 plants-11-03394-f005:**
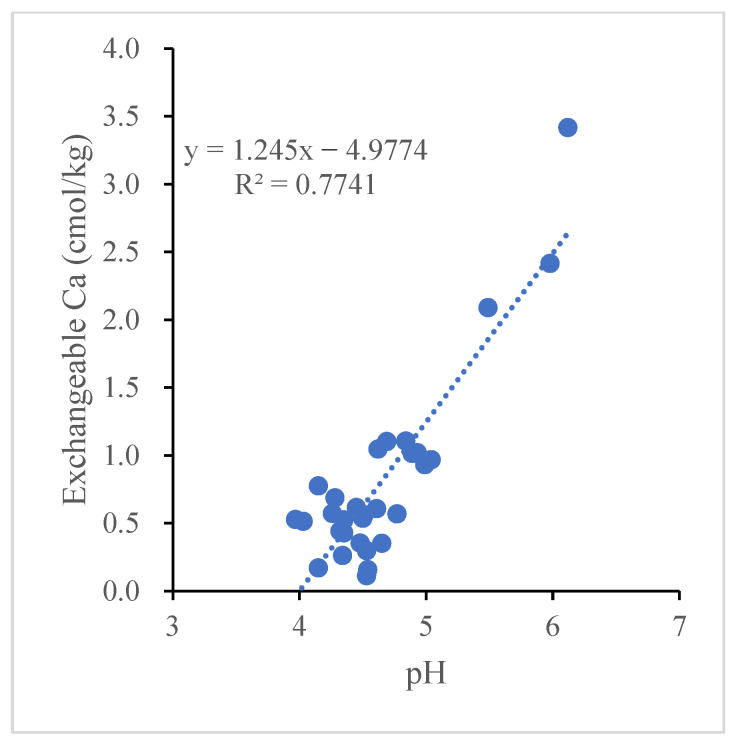
The correlation between the pH and exchangeable Ca in bayberry soils. Each dot represents a soil sample in the figure.

**Figure 6 plants-11-03394-f006:**
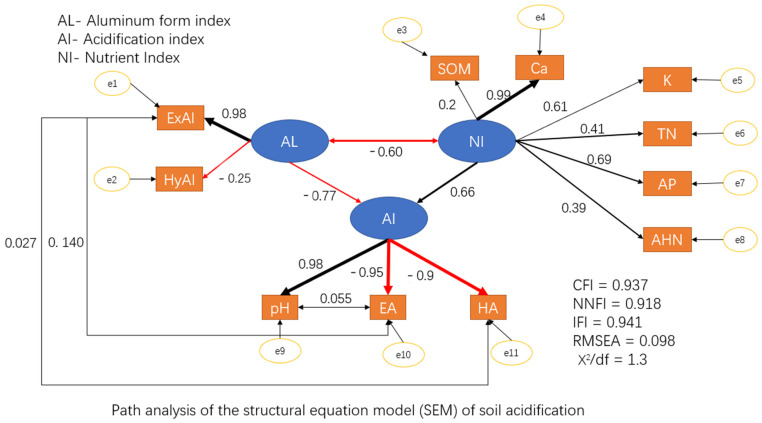
Correlation analysis of the soil parameters by the structural equation model. The thick solid lines indicate the significant correlation path (*p* < 0.01), and the thin lines indicate the nonsignificant correlation path. The number next to each line is the standardized path coefficient, and the width of an arrow indicates the strength of the relationship.

**Table 1 plants-11-03394-t001:** Assessment of bayberry decline disease by incidence and disease index on leaves and single fruit weight.

The Disease Assessment at Study Sites			SY ^a^	CX	DH	LX	QT	XJ	LH	TT	TS	WC
10-year-old trees	incidence and severity	percent of deciduous leaf (%)	42.3	25.6	39.6	41.7	35.4	16.3	35.8	37.8	36.0	33.6
		disease index	30.2	12.5	28.5	29.6	23.3	23.9	22.7	25.7	22.9	22.5
	incidence and severity	percent fruit weight (%)	40.3	22.6	36.6	40.7	32.4	14.0	33.8	34.8	31.0	31.6
		disease index	29.6	15.9	26.9	27.0	20.7	23.3	22.1	26.1	21.3	22.9
20-year-old trees	incidence and severity	percent of deciduous leaf (%)	51.7	36.0	48.0	51.1	43.8	26.4	46.2	47.2	42.4	43.0
		disease index	45.2	27.5	43.5	43.6	36.3	18.5	35.7	41.7	36.9	35.5
	incidence and severity	percent fruit weight (%)	52.3	35.6	48.6	52.7	46.4	27.0	45.8	45.8	43.0	43.6
		disease index	37.6	18.9	34.9	36.9	28.7	11.3	30.1	34.1	27.3	25.9
30-year-old trees	incidence and severity	percent of deciduous leaf (%)	59.5	43.6	55.6	56.7	52.4	35.0	52.8	54.8	50.0	48.6
		disease index	55.2	37.5	52.5	52.6	47.3	27.9	48.7	53.7	42.9	47.5
	incidence and severity	percent fruit weight (%)	65.3	45.6	61.6	64.7	55.4	37.0	55.8	61.5	56.0	54.6
		disease index	43.6	25.9	40.9	43.0	34.7	17.3	33.1	41.1	35.3	34.9

^a^ The abbreviations for county names of the study site: SY for Shangyu, CX for Cixi, DH for Dinghai, LX for Lanxi, QT for Qingtian, XJ for Xianju, LH for Linhai, TT for Tiantai, TS for Taishun, and WC for Wencheng.

**Table 2 plants-11-03394-t002:** Correlation coefficient between the soil cations (H^+^ and aluminum) and soil nutrients.

Soil Nutrient Factors	pH	Exchangeable Acidity	Hydrolytic Acidity	Exchangeable Aluminum
Soil organic matter	−0.0703 ns	−0.0263 ns	0.0280 ns	−0.0450 ns
Total carbon	−0.1536 ns	−0.0729 ns	0.1094 ns	−0.1223 Ns
Available phosphorus	0.4343 ns	−0.5018 *	−0.3414 ns	−0.5052 *
Total phosphorus	0.7490 ***	−0.6509 ***	−0.5147 *	−0.6234 **
Available nitrogen	0.2155 ns	−0.1789 ns	−0.1282 ns	−0.2141 ns
Total nitrogen	0.3626 *	−0.5047 *	−0.3723 ns	−0.4928 *
Total potassium	0.0929 ns	−0.1059 ns	0.0294 ns	−0.0225 ns
Total iron	−0.009 ns	0.0986 ns	0.0298 ns	0.2353 ns
Total calcium	0.344 ns	−0.3218 ns	−0.3622 ns	−0.3125 ns

The notes for the asterisks and the abbreviation ns in the table: * significant level at 0.05, ** significant level at 0.01, *** significant level at 0.001, and ns: no significance.

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
