# Peer review of "Investigation and Analysis of Rhizosphere Soil of Bayberry-Decline-Disease Plants in China"

_plants, 2022, doi:10.3390/plants11233394_

Round 1
Reviewer 1 Report
Overall Comments
- the study is well described and well conducted and presented
- the results are relevent
- the discussion of the relation between age of tree/root and soil parameters might be formulated a bit more explicitly regarding what is known about which was the cause and which was the effect.
- if possible recommendations about possible treatments of the plot (fertilizing or other measures) might complement the study.
Detailed Comments
Abstract:
- replace 'with 10, 20, and 30 years old' with 'with an age of 10, 20, and 30 years'
- replace 'K+ has fell' with 'K+ has fallen'
- replace 'more searious disease trees' with 'trees more seriously affected by the disease'
- replace 'accompagned lower' with 'accompagned with lower'
- replace 'shew' with 'have shown'
1. Introduction
- replace 'is signed' with 'is cited as' or 'is named as'
- replace 'divided' with 'seperated'
- upward shift '+' in Al3+'
- replace 'on survey' with 'on a survey'
Discussion
- replace 'has fell' with 'has fallen' or with 'fell'
Author Response
Author’s response to two reviewers (referees)
Firstly, thank you very much for the good advice. We are very glad to accept the suggestions and make revisions to improve the manuscript. It is necessary to know that some revision cannot be seen due to large changes by the addition or deletion or adjustment of the contents.
Reviewer 1
(Total ten comments from review 1)
Comments and Suggestions for Authors
Overall Comments
- the study is well described and well conducted and presented
- the results are relevent
- the discussion of the relation between age of tree/root and soil parameters might be formulated a bit more explicitly regarding what is known about which was the cause and which was the effect.
- if possible recommendations about possible treatments of the plot (fertilizing or other measures) might complement the study.
Detailed Comments
Author’s response: Thank you very much for above comments.
Abstract (total five comments from review 1 in the abstract)
Reviewer 1’ comment 1 of 10: - replace 'with 10, 20, and 30 years old' with 'with an age of 10, 20, and 30 years'
Author’s response: Thank you very much for the comment. We are very glad to accept the suggestions and replace 'with 10, 20, and 30 years old' with 'with an age of 10, 20, and 30 years'. Please to see the part of abstract in the manuscript for more detail.
Reviewer 1’ comment 2 of 10 - replace 'K+ has fell' with 'K+ has fallen'
Author’s response: Thank you very much for the comment. We are very glad to accept the suggestions and replace 'K+ has fell' with 'K+ has fallen'
Reviewer 1’ comment 3 of 10 - replace 'more serious disease trees' with 'trees more seriously affected by the disease'
Author’s response: Thank you very much for the comment. We are very glad to accept the suggestions and replace 'more serious disease trees' with 'trees more seriously affected by the disease'
Reviewer 1’ comment 4 of 10 - replace ' accompanied lower' with ' accompanied with lower'
Author’s response: Thank you very much for the comment. We are very glad to accept the suggestions and replace ' accompanied lower ' with 'accompanied with lower'
Reviewer 1’ comment 5 of 10 - replace 'shew' with 'have shown'
Author’s response: Thank you very much for the comment. We are very glad to accept the suggestions and
- Introduction (total four comments from review 1 in the Introduction)
Reviewer 1’ comment 6 of 10 - replace 'is signed' with 'is cited as' or 'is named as'
Author’s response: Thank you very much for the comment. We are very glad to accept the suggestions and replace 'is signed' with 'is cited as' or 'is named as'
Reviewer 1’ comment 7 of 10 - replace 'divided' with 'seperated'
Author’s response: Thank you very much for the comment. Because of contents adjustment, the sentence where the ‘divided’ belong to was deleted.
Reviewer 1’ comment 8 of 10 - upward shift '+' in Al3+'
Author’s response: Thank you very much for the comment. We are very glad to accept the suggestions and upward shift '+' in Al3+'.
Reviewer 1’ comment 9 of 10 - replace 'on survey' with 'on a survey'
Author’s response: Thank you very much for the comment. We are very glad to accept the suggestions and replace 'on survey' with 'on a survey' shown with marker in yellow Highlighted background.
Discussion (total one comment from review 1 in the Discussion)
Reviewer 1’ comment 10 of 10 - replace 'has fell' with 'has fallen' or with 'fell'
Author’s response: Thank you very much for the comment. We are very glad to accept the suggestions and 'has fell' with 'has fallen', at the same time, other same problem was corrected and shown with marker in yellow highlighted background.
Reviewer 2 Report
Interesting work but it needs major revision.
Title
Misleading there is no data on the occurrence of the disease in the paper. What’s more the diseased trees affected the soil or soil properties affected the trees, causing the disease?
Introduction
Very short, it does not sufficiently cover important topics such as: possible factors that may cause the disease, methods of control, why only aluminum was discussed?
“The public data showed that soil acidification amendments could effectively alleviate bayberry decline disease [3,4]” it is not discussed – later authors suggest that low pH of soil may promote the disease – it's contradictory.
“Our objectives were to discover the effects of decline diseased bayberry on soil parameters, to explore possible relations between the change of soil parameters and the disease.” What author's mean? Effect of disease on soil parameters? It is not clear.
Results
There is no information on the degree of plant damage by the disease. Information about the age of the plantation is not enough.
Discussion
The results are not well discussed.
Materials and Methods
Why is the material and methods chapter at the end of the work?
‘…the disease trees…” – what does it mean? – it should be “diseased trees”
The description of research stands is insufficient. It is not known whether trees of different ages were present in every orchard or only in selected ones?
What is the control? Were healthy trees/orchards evaluated and compared with affected by the disease.
There is no information on the assessment of the health condition of trees - without it, the work does not make sense. It cannot be assumed that the older orchard is more affected by the disease. Results from this range must be presented!
Conclusions
“Higher concentration of exchangeable aluminum was observed in the soil with more serious disease trees, …” – but there is no data related to plants infestation with the disease.
“Overall, these findings provided a new insight to mitigate bayberry decline disease by regulating these soil parameters.” – what does it exactly means?
Author Response
Author’s response to two reviewers (referees)
Firstly, thank you very much for the good advice. We are very glad to accept the suggestions and make revisions to improve the manuscript. It is necessary to know that some revision cannot be seen due to large changes by the addition or deletion or adjustment of the contents.
Reviewer 2
(Total six comments from review 2)
Comments and Suggestions for Authors
Interesting work but it needs major revision.
Author’s response: Firstly, thank you very much for the good advice. We are very glad to accept the suggestions and make corresponding revisions to improve the manuscript. Although answering those comments are labor-consuming and time-consuming jobs, essential improvement was obtained for the manuscript. Hence, thanks again.
Title
Reviewer 2’ comment 1 of 6: Misleading there is no data on the occurrence of the disease in the paper. What’s more the diseased trees affected the soil or soil properties affected the trees, causing the disease?
Author’s response: Firstly, thank you very much for the good advice. We are very glad to accept the suggestions and replace the old title with more proper title “Investigation and analysis rhizosphere soil of bayberry-decline-disease plants in China”
Introduction
Reviewer 2’ comment 2 of 6: Very short, it does not sufficiently cover important topics such as: possible factors that may cause the disease, methods of control, why only aluminum was discussed?
“The public data showed that soil acidification amendments could effectively alleviate bayberry decline disease [3,4]” it is not discussed – later authors suggest that low pH of soil may promote the disease – it's contradictory.
“Our objectives were to discover the effects of decline diseased bayberry on soil parameters, to explore possible relations between the change of soil parameters and the disease.” What author's mean? Effect of disease on soil parameters? It is not clear.
Author’s response: Firstly, thank you very much for the good advice.
We are very glad to accept the suggestions and add the contents about cause the disease and methods of control to avoid the short of the introduction, and add the content about ‘low pH of soil may promote the disease’ to avoid the contradictory with “The public data showed that soil acidification amendments could effectively alleviate bayberry decline disease [3,4]”, and make the “Our objectives were to discover the effects of decline diseased bayberry on soil parameters, to explore possible relations between the change of soil parameters and the disease.” What author's mean? Effect of disease on soil parameters?” clear by rewrite the last paragraph.
Results
Reviewer 2’ comment 3 of 6: There is no information on the degree of plant damage by the disease. Information about the age of the plantation is not enough.
Author’s response: Firstly, thank you very much for the good advice. We added the contents about disease assessment and tree age. Please to see the manuscript.
Discussion
Reviewer 2’ comment 4 of 6: The results are not well discussed.
Author’s response: Firstly, thank you very much for the good advice. We rewrote the discussion to make it better, please to see the manuscript.
Materials and Methods
Reviewer 2’ comment 5 of 6: (1)Why is the material and methods chapter at the end of the work?
(2) ‘…the disease trees…” – what does it mean? – it should be “diseased trees”
(3) The description of research stands is insufficient. It is not known whether trees of different ages were present in every orchard or only in selected ones?
(4) What is the control? Were healthy trees/orchards evaluated and compared with affected by the disease.
(5) There is no information on the assessment of the health condition of trees - without it, the work does not make sense. It cannot be assumed that the older orchard is more affected by the disease. Results from this range must be presented!
Author’s response: Firstly, thank you very much for the good advice. (1) Because the template used in this manuscript come from office demands of MDPI, in which the material and methods chapter was required to place at the the end of the work. (2) All the “the disease trees” in the manuscript were replaced with “diseased trees”. (3) Different ages were present in every orchard. (4) Healthy trees are considered as the control, which were defined as no symptom caused by the disease on the trees. (5) The soil parameters in rhizosphere soil of healthy trees present no obvious rules compared the diseased trees. Hence, the information is still little, ever though some data was added in the new manuscript.
Conclusions
Reviewer 2’ comment 6 of 6: (1) “Higher concentration of exchangeable aluminum was observed in the soil with more serious disease trees, …” – but there is no data related to plants infestation with the disease.
“Overall, these findings provided a new insight to mitigate bayberry decline disease by regulating these soil parameters.” – what does it exactly means?
Author’s response: Firstly, thank you very much for the good advice. (1) The contents about disease assess was added in part of material s and method and Results. (2) For more clear description, the sentence “Overall, these findings provided a new insight to mitigate bayberry decline disease by regulating these soil parameters.” was changed into “Overall, these findings provided a new insight to mitigate bayberry decline disease by regulating above changed soil parameters.”
Round 2
Reviewer 2 Report
The manuscript has been sufficiently improved.